# Opioid overdose and naloxone administration knowledge and perceived competency in a probability sample of Indiana urban communities with large Black populations

Shin Hyung Lee[1], Jon Agley[1], Vatsla Sharma[2], Francesca Williamson [3], Pengyue Zhang[4], Dong-Chul Seo [1]*

**1** Department of Applied Health Science, Indiana University School of Public Health, Bloomington, Indiana, United States of America, **2** O'Neill School of Public and Environmental Affairs, Indiana University-Bloomington, Bloomington, Indiana, United States of America, **3** Department of Learning Health Sciences, University of Michigan Medical School, Ann Arbor, Michigan, United States of America, **4** Department of Biostatistics and Health Data Science, Indiana University School of Medicine, Indianapolis, Indiana, United States of America

* seo@iu.edu

## Abstract

### Background

Opioid overdose deaths pose a serious public health concern in the United States, with disproportionately higher rates of increase among Black Americans despite expanded naloxone access. Improving community knowledge and confidence in naloxone use may be critical to reducing these disparities.

### Objective

This study assessed individual- and community-level factors associated with knowledge and perceived competency in managing opioid overdose and administering naloxone among urban Indiana residents.

### Methods

A probability-based household survey was conducted between March to May 2023 across eight Indiana zip code areas (N = 772) with high (> 40%) proportions of Black residents. Multilevel modeling was used to examine individual- and community-level factors associated with opioid overdose knowledge and perceived competency, using adapted items from the Opioid Overdose Knowledge Scale (OOKS) and Opioid Overdose Attitudes Scale (OOAS).

**Data availability statement:** The datasets generated and/or analyzed during the current study are not publicly available due to the inclusion of personally identifiable information and sensitive data on illicit drug use collected from small, specific geographical areas (eight ZIP code areas). However, data may be made available upon reasonable request. Requests should be submitted to the Human Research Protection Program (HRPP), Office for Research Compliance, Indiana University, 509 E. Third St., Bloomington, Indiana 47401, USA (Email: irb@iu.edu). Upon receiving a request, the HRPP will work with the study investigators to deidentify the dataset by removing ZIP code and race/ethnicity information, and will seek approval from the Indiana University Institutional Review Board to ensure the appropriateness of the deidentification process before any data are shared.

**Funding:** This Project was supported by the Office of Minority Health (OMH) of the U.S. Department of Health and Human Services (HHS) as part of a financial assistance award (CPIMP221346). The contents are those of the authors and do not necessarily represent the official views of, nor an endorsement, by OMH/OASH/HHS, or the U.S. Government. For more information, please visit https://www.minori-tyhealth.hhs.gov/. There was no additional external funding received for this study.

**Competing interests:** The authors have declared that no competing interests exist.

## Results

Race, sex, household income, education, time lived in the community, and history of opioid overdose significantly predicted knowledge scores. White participants scored higher (mean 6.65) than Black participants (5.70) ($p < 0.001$). A significant cross-level interaction was found, with Black residents living in high-poverty areas scoring lower than White counterparts ($\beta = 1.06$, $p = 0.039$). In contrast, perceived competency was primarily associated with age and personal history of overdose.

## Conclusions

Racial and socioeconomic disparities persist in opioid overdose knowledge, particularly among Black residents in low-income communities. Tailored, culturally responsive education efforts, especially by trusted community members, may help improve overdose response readiness.

## Introduction

With the recent proliferation of illicit synthetic opioids, including fentanyl and its analogues, Indiana, like other states, is facing a significant public health challenge from opioid-involved overdoses [1,2]. In 2022, Indiana ranked as the 13th highest state in drug overdose mortality nationwide, with a rate of 41.0 deaths per 100,000 population. Moreover, Indiana recorded 2,072 deaths attributed to opioid overdoses, comprising 77% of the overall drug overdose fatalities within the state in 2022 [3]. In Indianapolis, the most populous city in Indiana, there were 195 fentanyl-related drug overdose deaths in 2018, and this number had increased to 799 deaths in 2021 [1].

Researchers have expressed concern that the increase in opioid-involved overdoses and fatalities is disproportionately affecting marginalized communities, with Black Americans in urban settings particularly impacted [4–6]. In 2018 and 2019, the age-adjusted opioid overdose death rate per 100,000 people was higher for White people (18.8 and 19.2, respectively) than for Black people (14.1 and 17.3, respectively), but in 2020, data began showing a higher death rate for Black people than White people and diverging trend lines. By 2022, the opioid [7] overdose death rate for Black people was 36.6 and the rate for White people was 27.6 [3]. Gondré-Lewis et al. (2023) also reported that several major urban areas, including Baltimore, Chicago, Detroit, and Philadelphia, with large Black populations, had markedly higher opioid-related death rates compared to rural areas [5]. Similarly, Black communities in Indiana experienced the largest increase in opioid-related overdose deaths in recent years [8]. However, the mechanisms behind these racial differences for prevalence rates and trends, especially in urban Indiana settings, remain poorly understood.

Administration of naloxone to reverse opioid overdoses is one key area where data suggest population-level differences that may be associated with opioid-related death rates. Naloxone, an opioid antagonist, is a safe and effective medication that can be easily administered to anyone undergoing an opioid overdose to reverse that

overdose [9,10]. Research suggests that the incidence of fatal opioid overdoses could decrease if community members have access to naloxone and are educated about opioid-involved overdose and naloxone administration [11,12]. Yet, in their Baltimore city based study, Dayton et al. (2020) found that White participants had significantly higher odds of receiving naloxone training and using naloxone compared to Black participants [13]. In their study on drug overdose deaths, Takemoto et al. (2024) and Ray et al. (2019) found racial differences in naloxone administration, with White decedents more likely to have had naloxone administered compared to Black decedents [14,15]. At the same time, most U.S. states have enacted legislation to expand its accessibility and utilization among laypersons to address public health emergency of opioid-involved fatalities [9,16,17]. In addition, the U.S. Food and Drug Administration recently approved some formulations of naloxone for over-the-counter sale nationwide [18].

The State of Indiana has undertaken substantial efforts to expand access to naloxone in recent years, and more than 578,000 doses of naloxone have been distributed throughout Indiana as part of these efforts since 2020 [19]. Still, there are multiple factors that may hinder timely naloxone administration at overdose scenes, such as lack of knowledge about opioid overdose and naloxone administration methods [20]. Such barriers to timely naloxone administration to reverse opioid-involved overdoses are complex [20] and may be related to insufficient understanding of opioid overdose and naloxone administration [21,22]. Courtney et al. (2023) found that lack of knowledge about opioid overdose and naloxone administration among laypersons were among the major barriers for using naloxone at overdose scenes [21].

Despite the importance of reducing rates of opioid-involved overdoses, no prior studies have specifically examined both individual- and community-level factors shaping opioid overdose knowledge and perceived naloxone competency in Black urban communities. Existing research has primarily focused on disparities in naloxone access, fatality trends or training opportunities, but not sufficiently explored how structural inequities intersect with individual knowledge and perceived readiness to intervene at an overdose encounter particularly in urban areas. In order to develop effective interventions, it is necessary to better understand the individual- and community-level factors associated with opioid overdose knowledge and perceived naloxone competency. Besides, this study addresses the gap by using recent probability samples across multiple Indiana cities with high proportions of Black residents to identify disparities in overdose-related knowledge and naloxone competency. By incorporating both individual-level and neighborhood-level characteristics, we aim to provide baseline data critical for informing community-based interventions designed to better ensure that all communities have access to effective means of reducing opioid-involved overdose fatality rates.

Accordingly, the objectives of this study were (1) to measure the level of knowledge and perceived competency regarding opioid overdose and naloxone administration in recent probability samples of Black urban communities in Indiana, (2) investigate racial and other sociodemographic correlates of knowledge and perceived competency, and (3) assess possible relationships, if any, between community-level factors (e.g., proportion of Black residents in the community, community-level income and employment rates) and knowledge and perceived competency.

## Methods

### Ethics statement

This study was approved by the Indiana University Institutional Review Board (IUIRB), which determined that the research met the federal and university criteria for exemption (approved protocol ID: 16953). Informed consent was obtained for all participants. Waiver of documentation of consent was approved by the IUIRB.

### Sample and administration

This study was part of the Multi-Sector and Multi-Level Community-Driven Approaches to Remove Structural Racism and Overdose Deaths in Black Indianapolis Communities (MACRO-B), a project that includes opioid overdose and naloxone administration community education programming to reduce opioid overdose deaths in Black Indianapolis communities.

As part of the project, data were collected from March 2023 to May 2023 via mail and web-based household probability community surveys with samples of non-institutionalized adults aged 18 years or older. Demographic characteristics of survey respondents are shown in S1 Table. Data collection was managed by the Indiana University Center for Survey Research. Probability sampling was used to select respondents representative of eight urban zip code areas in Indiana with Black populations greater than 40%. Four zip code areas in inner-city Indianapolis (46202, 46205, 46208, and 46218) were selected as the target intervention communities due to their high proportion of Black residents and elevated rates of opioid overdose deaths compared to other areas in Indiana. To have a rigorous evaluation of intervention outcomes and minimize potential confounding, four additional zip code areas in northern Indiana, Gary (46408), Merrillville (46410), South Bend (46628), and Fort Wayne (46806), were selected as comparison communities. These areas were matched to the target communities using zip-code level data on four key criteria: the proportion of Black residents, the percentages of families living in poverty, overall population size, and the proportion of school-aged children eligible for free or reduced lunch. Matching data were obtained from the United States Census Bureau and Indiana State Department of Health over-dose surveillance reports [7,19]. The intervention and comparison communities were similar in terms of the four matching variables while they were geographically distant (i.e., 125–150 miles away), helping minimize contamination due to diffusion of treatment while maintaining similarities in key sociodemographic characteristics.

The study adhered to the guidelines established by the American Association for Public Opinion Research (AAPOR) [23]. Survey administration proceeded in sequential stages. On day 1, households were mailed a preliminary letter containing a $1 pre-incentive and instructions for completing an online survey, along with notification that a $20 gift card would be provided upon completion. Reminder letters were sent to non-respondents on day 10 and day 20, with the second reminder including a paper version of the survey and a self-addressed, stamped return envelope. A third reminder with a paper version of the survey was mailed on day 35, followed by a final reminder on days 50–55, encouraging participation either online or by mail. The overall response rate was 23%, calculated using AAPOR Response Rate Formula #4 [23]. A detailed summary of the survey administration timeline is available in S1 Fig.

## Measures

**Knowledge on opioid overdose and naloxone administration.** To assess knowledge, 10 items assessing opioid overdose knowledge were adapted from the Opioid Overdose Knowledge Scale (OOKS) [24]. In the current sample, the internal consistency reliability of the 10-item knowledge scale was modest (Cronbach's alpha = 0.41). This is consistent with prior research suggesting that lower Cronbach's alpha values are acceptable for knowledge-based assessments, particularly when measuring diverse domains of factual information [25,26]. The questionnaire included 6 true/false items and 4 multiple-choice items designed to test basic knowledge about identifying the risks, signs, and symptoms of opioid overdose, as well as understanding the proper use of naloxone (see S2 Table for these items). Each correct answer earned one point, resulting in a possible score range of 0–10 for knowledge on opioid overdose and naloxone administration. We calculated mean scores by race, ensuring that all scores were weighted accordingly.

**Perceived competency to manage opioid overdoses.** To assess the perceived level of competency in managing opioid overdoses, we asked the survey respondents to rate their agreement with 7 items on a Likert-type scale ranging from 1 to 5 (see S2 Table for these items). These items were also derived from the Opioid Overdose Knowledge Scale (OOKS) and the Opioid Overdose Attitudes Scale (OOAS) [24]. Each item was scored individually, and appropriate reverse coding was made so that a higher score always indicated a higher perceived level of competency (Cronbach's alpha = 0.87). We then calculated the weighted mean scores of all 7 items by race.

**Individual-level and zip code-level predictors.** One of the key variables in this study was race. In analyses, variable responses were dummy coded as White, Black, and Other. Due to the small sample sizes for specific groups (Asian = 22, American Indian or Alaska Native = 21, and Native Hawaiian or Pacific Islander = 6), we combined those individuals with the "Other" category to ensure sufficient numbers for meaningful analyses [27]. Geographic area was

determined by the eight zip code areas, which were then dummy coded by region as inner-city Indianapolis (46202, 46205, 46208, and 46218) and other urban areas such as Gary (46408), Merrillville (46410), South Bend (46628), and Fort Wayne (46806). Several demographic factors were included as covariates: age group (18–24, 25–34, 35–44, 45–54, 55–64, 65–74, and 75+), biological sex at birth (women, men), ethnicity (Non-Latino, Latino), educational level (some college or higher, high school/GED or less), and household income, which had the following categories: less than $15,000; $15,000–$24,999; $25,000–$34,999; $35,000–$49,999; $50,000–$74,999; $75,000–$99,999; and $100,000 or more. For analysis purposes, income was collapsed into three broader categories (over $100,000, $35,000-$99,999, less than $35,000). Other covariates included duration of residence in the community (more than 3 years, 1–3 years, less than a year) and history with opioid overdose (yes/no). The variable for opioid overdose history was created based on responses to two survey questions. A respondent who reported yes to either (1) personally experiencing an opioid overdose or (2) having a familiy member or close friend who died from an opioid overdose was coded as yes and otherwise no. Additionally, zip code-level factors were included such as proportion of Black residents, number of individuals with income below the 200 percent of federal poverty level (FPL<200%), median household income, proportion of individuals with high school or higher educational attainment among those who were 25 years or older, and employment rate. These zip-code level data are shown in S3 Table. These factors were obtained from the 2022 American Community Survey 5-year estimates [7].

### Statistical analysis

Out of the 772 survey respondents, 47 cases with missing information were excluded using listwise deletion, resulting in a final analytic sample of 725 respondents. Given the low rate of missingness 6% [28], listwise deletion was appropriate to maintain a consistent analytic sample across variables [29]. Descriptive statistics included unweighted frequencies and weighted proportions for demographic characteristics and correct responses on opioid overdose and naloxone administration questions. Weighted mean scores were calculated for perceived competency items. Categorical variables were compared across racial groups using adjusted chi-square tests with design-based Rao-Scott corrections. Continuous variables were compared using design-based F-tests, accounting for the complex survey design.

Multilevel analysis models were then constructed to identify factors associated with two outcomes: (a) knowledge and (b) perceived competency regarding opioid overdose and naloxone administration. Following Lee et al. (2015), nested models were sequentially built, beginning with a null model, then adding individual-level socioeconomic variables, overdose experience, zip code-level predictors, and cross-level interactions [30]. All analyses were conducted using R version 4.3.2 [31].

## Results

### Demographic characteristics

Table 1 provides a breakdown of the demographic and socioeconomic characteristics of the survey respondents categorized by race, including White (n = 359), Black (n = 347), and Other racial groups (n = 60). There were significant differences by race observed for the following variables: ethnicity ($p = 0.002$), age group ($p < 0.001$), educational attainment ($p = 0.001$), household income ($p < 0.001$), and opioid overdose history ($p = 0.003$). Region, sex, and length of time lived in the communities were not significantly different by race.

### Weighted scores for knowledge on opioid overdose and naloxone administration and perceived competency level to manage opioid overdoses by race

Table 2 presents the unweighted frequencies and weighted proportions of responses to the 10 questions on opioid overdose and naloxone administration knowledge, along with the weighted mean scores of 7 items that assess perceived

**Table 1. Demographic characteristics by Race: March-May 2023, (N = 772).**

| Variable | White (n = 359) | Black (n = 347) | Other (n = 60) | p-value |
|---|---|---|---|---|
| | n (%) | n (%) | n (%) | |
| Ethnicity | | | | 0.002 |
| Non-Latine | 323 (87.5) | 330 (95.6) | 43 (76.1) | |
| Latine | 35 (12.5) | 13 (4.4) | 16 (23.9) | |
| Age group | | | | <0.001 |
| 18-24 years | 38 (14.7) | 27 (15.2) | 11 (22.9) | |
| 25-34 years | 82 (30.3) | 36 (9.6) | 10 (16.0) | |
| 35-44 years | 63 (16.6) | 54 (14.4) | 16 (31.8) | |
| 45-54 years | 39 (8.9) | 54 (16.6) | 9 (10.0) | |
| 55-64 years | 47 (12.3) | 67 (22.3) | 6 (6.2) | |
| 65-74 years | 56 (10.6) | 71 (14.7) | 5 (9.6) | |
| 75 years or older | 33 (6.6) | 38 (7.2) | 2 (3.5) | |
| Biological sex at birth | | | | 0.600 |
| Women | 217 (55.0) | 247 (59.2) | 41 (63.7) | |
| Men | 142 (45.0) | 99 (40.8) | 19 (36.3) | |
| Educational attainment | | | | 0.001 |
| Some college or higher | 288 (67.0) | 250 (47.3) | 48 (73.5) | |
| High school/GED or less | 71 (33.0) | 97 (52.7) | 12 (26.5) | |
| Household income | | | | <0.001 |
| Upper, more than $100,000 | 94 (29.8) | 46 (11.5) | 14 (27.3) | |
| Middle, $35,000 - $99,999 | 161 (46.2) | 151 (33.8) | 21 (33.3) | |
| Lower, less than $35,000 | 99 (29.8) | 148 (54.7) | 23 (39.4) | |
| Length of time lived in the communities | | | | 0.554 |
| More than three years | 244 (65.6) | 261 (72.7) | 41 (70.1) | |
| One year to three years | 83 (27.3) | 66 (20.1) | 15 (25.7) | |
| Less than a year | 32 (7.1) | 19 (7.2) | 4 (4.2) | |
| Opioid overdose history[a] | | | | 0.003 |
| No | 264 (69.1) | 275 (78.4) | 53 (93.7) | |
| Yes | 95 (30.9) | 72 (21.6) | 6 (6.3) | |
| Region | | | | 0.996 |
| Indianapolis | 204 (54.3) | 175 (54.8) | 40 (54.8) | |
| Other | 155 (45.7) | 172 (45.2) | 20 (45.2) | |

*Notes.* Frequencies are unweighted, and proportions are weighted. GED = General Educational Development. Differences between groups were examined via weighted adjusted Chi-Square tests.

[a]Opioid overdose history was constructed from responses to two survey items: (1) whether the respondent had personally experienced an opioid overdose and (2) whether a family member or close friend had died from an opioid overdose. A respondent who reported yes to either question was coded as yes.

competency in managing opioid overdose. The average score on the 10 questions assessing knowledge on opioid overdose and naloxone administration showed significant differences by race ($p < .001$). White participants had the highest scores (Mean = 6.65, $SD$ = 1.86), followed by other racial groups (Mean = 5.75, $SD$ = 1.89), and then Black participants (Mean = 5.70, $SD$ = 1.79). In contrast, perceived competency to manage opioid overdose, assessed through 7 survey items, did not show significant variation across racial groups ($p$ = 0.478). White participants had a mean score of 2.95 ($SD$ = 1.05), Black participants scored 2.82 ($SD$ = 0.92), and Other racial groups scored 2.96 ($SD$ = 1.07).

Table 2. **Weighted scores for knowledge on opioid overdose and naloxone administration and perceived competency level to manage opioid overdoses by Race: March-May, 2023 (N = 772).**

| | White (n = 359) | Black (n = 347) | Other (n = 60) | p-value[a] |
|---|---|---|---|---|
| | | Mean (SD) | | |
| *Knowledge on Opioid Overdose and Naloxone Administration (10 Questions)* | 6.65 (1.86) | 5.70 (1.79) | 5.75 (1.89) | <0.001 |
| *Perceived Competency to Manage Opioid Overdose (7 items)* | 2.95 (1.05) | 2.82 (0.92) | 2.96 (1.07) | 0.478 |

*Notes.* Frequencies may not sum to the total due to missing observations.

[a]Corresponding p-values were calculated via weighted design-based F-tests.

Some individual knowledge questions (components of the overall knowledge score) showed significant variation by racial group, whereas none of the individual weighted mean scores for perceived competence items (components of the overall weighted mean competency score) indicated significant differences among racial groups.

## Factors associated with knowledge on opioid overdose and naloxone administration scores

Table 3 presents the results of a multilevel model assessing individual- and zip code- level predictors of knowledge about opioid overdose and naloxone administration. Several individual-level characteristics were significantly associated with knowledge scores. In the model, Black (β = −1.74, p = 0.005) participants had significantly lower opioid overdose and naloxone administration knowledge scores compared to their White counterparts. Education level also played a significant role, with participants having a high school education or less scoring lower than those with some college or higher education ($\beta = -0.31$, $p = 0.044$). Income was another significant predictor; compared to those earning > $100,000, those earning < $100,000 showed lower knowledge scores ($\beta = -0.47$, $p = 0.009$ for those earning $35,000 – $99,999 and $\beta = -0.66$, $p = 0.002$ for those earning less than $35,000). Notably, participants with a history of opioid overdose demonstrated higher knowledge scores ($\beta = 0.83$, $p = <0.001$). For the length of time lived in the community, individuals who lived in their area for one to three years scored lower than those who lived there for more than three years ($\beta = -0.42$, $p = 0.012$), possibly reflecting differences in familiarity with local resources. Zip code-level predictors showed no significant relationship with knowledge scores. However, a significant cross-level interaction was found between Black race and FPL < 200%, indicating that Black residents living in poorer neighborhoods displayed lower knowledge scores than White counterparts ($\beta = 1.06$, $p = 0.039$). The intraclass correlation coefficient (ICC) for the null model was 0.028, indicating that 2.8% of the total variance in the knowledge level was due to differences between the zip codes. The findings suggest that knowledge on opioid overdose and naloxone administration may be shaped by both sociodemographic characteristics and neighborhood context.

## Factors associated with perceived competency on opioid overdose and naloxone administration

Table 4 summarizes the multilevel model analysis on perceived competency in managing opioid overdose and administering naloxone. At the individual level, race was not a significant predictor, but age was. The youngest age group (18−24 years) reported significantly lower perceived competency ($\beta = -0.34$, $p = 0.013$) compared to the 35−44 age group, as did the oldest groups, aged 65−74 ($\beta = -0.45$, $p = 0.001$) and 75+ ($\beta = -0.33$, $p = 0.013$). No significant differences were noted for the other age groups. The only other significant predictor of perceived competency was having a personal or familial history of opioid overdose, which was associated with a higher score ($\beta = 0.61$, $p < 0.001$). The ICC of the null model was 0.004, indicating that only 0.4% of the variance in perceived competency was due to differences between zip codes. Random effects analysis showed minimal variability across zip codes. The findings suggest that age and lived experience with overdose may be key factors shaping confidence in administering naloxone.

**Table 3. Multilevel model analysis of knowledge on opioid overdose and naloxone administration (N = 723).**

| Variable | β | SE | *p*-value |
|---|---|---|---|
| **Fixed Effects** | | | |
| *Individual-level predictors* | | | |
| Intercept | | | |
| Race | | | |
| White | Ref | | |
| Black | **−1.74**** | 0.62 | 0.005 |
| Other | −1.02 | 0.89 | 0.252 |
| Ethnicity | | | |
| Non-Latino | Ref | | |
| Latino | −0.40 | 0.24 | 0.102 |
| Age group | | | |
| 18-24 years | 0.39 | 0.24 | 0.109 |
| 25-34 years | 0.38 | 0.22 | 0.086 |
| 35-44 years | Ref | | |
| 45-54 years | −0.01 | 0.24 | 0.964 |
| 55-64 years | 0.11 | 0.24 | 0.634 |
| 65-74 years | 0.26 | 0.25 | 0.309 |
| 75 years or older | −0.07 | 0.31 | 0.813 |
| Biological sex at birth | | | |
| Women | Ref | | |
| Men | **−0.36**** | 0.13 | 0.007 |
| Educational attainment | | | |
| Some college or higher | Ref | | |
| High school/GED or less | **−0.31*** | 0.16 | 0.044 |
| Household income | | | |
| Upper, more than $100,000 | Ref | | |
| Middle, $35,000 - $99,999 | **−0.47**** | 0.18 | 0.009 |
| Lower, less than $35,000 | **−0.66**** | 0.21 | 0.002 |
| Length of time lived in the communities | | | |
| More than three years | Ref | | |
| One year to three years | **−0.42*** | 0.17 | 0.012 |
| Less than a year | −0.37 | 0.29 | 0.206 |
| Opioid overdose history[a] | | | |
| Yes | **0.83***** | 0.16 | <0.001 |
| No | Ref | | |
| Region | | | |
| Indianapolis | 0.33 | 0.26 | 0.206 |
| Other | Ref | | |
| *Zip code-level predictors* | | | |
| Black proportion | −1.04 | 1.43 | 0.467 |
| FPL<200%[c] | −0.26 | 0.71 | 0.715 |
| *Cross-level interaction* | | | |
| Race (Black) * FPL<200% | **1.06*** | 0.51 | 0.039 |

*(Continued)*

**Table 3.** (Continued)

| Variable | β | SE | *p*-value |
|---|---|---|---|
| **Random Effects** | | | 95% CI[b] |
| τ₀₀ (Level-2) | | 0.26 | 0.01–0.35 |
| σ² (Level-1) | | 1.70 | 1.60–1.77 |

*Note.* Ref = reference group. GED = General Educational Development. FPL = Federal Poverty Level. Due to missing data on the variables, the analytical sample was reduced to 723.

[a] Opioid overdose history include individuals with personal opioid overdose experience as well as experiences involving the death of a family member or close friend due to opioid overdose.

[b] 95% profile confidence interval

[c] The number of individuals for FPL < 200% was divided by 10,000 to reduce its scale in analysis.

\* *p* < .05 \*\* *p* < .01 \*\*\* *p* < .001.

## Discussion

This study is one of the first to examine knowledge and perceived competency regarding opioid overdose and naloxone administration among racially diverse urban residents in Indiana. This study contributes to the growing literature on racial and socioeconomic disparities in opioid overodse responses by focusing on knowledge and perceived competency among urban residents in Indiana. Although previous research has documented increasing overdose deaths among Black Americans nationally, less is known about how knowledge and readiness to respond to overdoses vary by race and neighborhood conditions at the local level. By using probability samples across major urban areas of Indiana, our study provides novel baseline data that highlights significant differences. White urban residents scored the highest on knowledge items, while Black residents scored the lowest. The lower knowledge scores observed among Black residents, particularly those living in high-poverty neighborhoods, may reflect the combined impact of social disadvantage and neighborhood-level resource gaps. These findings reveal important structural and contextual barriers to overdose prevention, and suggest that future overdose prevention efforts that are specifically tailored by race and socioeconomic status may help optimize prevention of fatal opioid-involved overdoses.

Our finding that Black urban residents scored lower on knowledge questions around opioid overdose and naloxone administration compared to White urban residents is consistent with prior research [32,33]. However, it is important to distinguish between disparities in knowledge or perceived competency and disparities in actual naloxone administration outcomes. Prior work has shown that although racial and ethnic disparities exist in who receives overdose education or naloxone administration training, no significant racial or ethnic differences were observed in survivorship following naloxone administration [32]. Nonetheless, the relatively low knowledge scores across all urban Indiana residents in the sample aligns with previous research indicating that despite naloxone's relatively widespread availability, knowledge about naloxone is lacking [22,34–36]. Our findings suggest that widespread distribution of naloxone may not guarantee good understanding on naloxone administration. This can be likely improved through peer-led educational interventions which have been shown to build trust and improve overdose response skills including naloxone administration [37–41].

Seo et al. (2023) found that Black residents in Indianapolis felt they lacked sufficient educational opportunities regarding opioid overdose and naloxone usage compared to White urban residents [42]. However, they expressed a readiness to learn about and carry naloxone for overdose response, particularly when the education and support came from trusted figures within their community such as Black church leaders [42]. Addressing the gap observed in this study might be achievable by implementing culturally tailored education programs [43,44] involving community leaders who grasp the cultural nuances and can communicate effectively with Black residents about the importance of naloxone administration [45].

Also consistent with previous studies [46,47], our findings indicated that having a history of opioid overdose, whether personal or through the death of a family member or close friend, was associated with significantly higher knowledge

**Table 4. Multilevel Model Analysis of Perceived Competency on Opioid Overdose and Naloxone Administration (N = 723).**

| Variable | β | SE | p-value |
|---|---|---|---|
| **Fixed Effects** | | | |
| *Individual-level predictors* | | | |
| Intercept | | | |
| Race | | | |
|   White | Ref | | |
|   Black | −0.58 | 0.29 | 0.093 |
|   Other[a] | 0.63 | 0.49 | 0.203 |
| Ethnicity | | | |
|   Non-Latine | Ref | | |
|   Latine | −0.25 | 0.13 | 0.068 |
| Age group | | | |
|   18-24 years | **−0.34*** | 0.13 | 0.013 |
|   25-34 years | −0.11 | 0.12 | 0.392 |
|   35-44 years | Ref | | |
|   45-54 years | −0.15 | 0.14 | 0.264 |
|   55-64 years | −0.04 | 0.13 | 0.764 |
|   65-74 years | **−0.45**** | 0.14 | 0.001 |
|   75 years or older | **−0.33*** | 0.17 | 0.013 |
| Biological sex at birth | | | |
|   Women | Ref | | |
|   Men | 0.05 | 0.07 | 0.480 |
| Educational attainment | | | |
|   Some college or higher | Ref | | |
|   High school/GED or less | −0.06 | 0.09 | 0.456 |
| Household income | | | |
|   Upper, more than $100,000 | Ref | | |
|   Middle, $35,000 - $99,999 | −0.08 | 0.10 | 0.429 |
|   Lower, less than $35,000 | −0.08 | 0.11 | 0.469 |
| Length of time lived in the communities | | | |
|   More than three years | Ref | | |
|   One year to three years | −0.02 | 0.09 | 0.817 |
|   Less than a year | −0.09 | 0.16 | 0.571 |
| Opioid overdose history[b] | | | |
|   Yes | **0.61***** | 0.09 | <0.001 |
|   No | Ref | | |
| Region | | | |
|   Indianapolis | −0.07 | 0.10 | 0.491 |
|   Other | Ref | | |
| *Zip code-level predictors* | | | |
|   Black proportion | 0.24 | 0.55 | 0.664 |
|   FPL<200%[c] | −0.08 | 0.31 | 0.878 |
| *Cross-level interaction* | | | |
|   Race (Black) * FPL<200% | 0.41 | 0.28 | 0.144 |

*(Continued)*

**Table 4.** (Continued)

| Variable | β | SE | *p*-value |
|---|---|---|---|
| **Random Effects** | | | 95% CI[b] |
| $\tau_{00}$ (Level-2) | | 0.00 | 0.00–0.00 |
| $\sigma^2$ (Level-1) | | 0.90 | 0.89–0.99 |

*Note.* Ref = reference group. GED = General Educational Development. FPL = Federal Poverty Level. Due to missing data on the variables, the analytical sample was reduced to 723.

[a] Opioid overdose history include individuals with personal opioid overdose experience as well as experiences involving the death of a family member or close friend due to opioid overdose.

[b] 95% profile confidence interval

[c] The number of individuals for FPL < 200% was divided by 10,000 to reduce its scale in analysis.

\* $p < .05$ \*\* $p < .01$ \*\*\* $p < .001$

scores. This suggests that firsthand experience or proximity to opioid-related events may serve as a catalyst for increased awareness and understanding [46,47], potentially due to greater engagement with medical professionals or support services following such events. Interestingly, no significant associations were found between zip code-level predictors and knowledge scores except the poverty level of the community. Research has shown that poverty and socioeconomic status are linked to educational access and health literacy [48] thereby affecting health outcomes [49]. Additionally, aligning with previous studies [47,50] several socioedemographic factors were significantly related with knowledge scores. Men, individuals with lower household income and educational attainment, and those in lower-income brackets exhibited lower knowledge scores. Strategies such as targeted education, improved access to resources, local partnerships, tailored messaging, financial incentives, and ongoing evaluation, may help address knowledge disparities among men, lower-income, and less-educated groups, though exploring their effectiveness lies beyond the scope of this study. Additionally, we found a cross-level interaction between race and poverty level. This interaction emphasizes the need for interventions that address both race and socioeconomic status such that more Black residents in high-poverty areas are provided with tailored educational opportunities on opioid overdose and naloxone administration.

Regarding perceived competency in managing opioid overdose and naloxone administration, age emerged as a significant predictor, with both the youngest (18–24 years) and oldest age groups (65–74 and 75 years or older) reporting lower perceived competency. This may indicate a lack of practical experience or exposure to training opportunities among these cohorts. Notably, having an opioid overdose history significantly enhanced perceived competency, aligning with our previous findings regarding knowledge scores. This relationship suggests that experiential learning may play an important role in developing confidence in overdose management. In a study based in Nebraska, Cooper-Ohm et al. (2023) found an insignificant relationship between past history and familiarity with naloxone in cities of Omaha and Lincoln, however a significant relationship was found in regions outside these two cities [51]. Furthermore, the ICC results from the null models in this study provide insights into the variability of knowledge and perceived competency related to opioid overdose management and naloxone administration across different urban communities. Specifically, the ICC showed that 2.8% of the variance in knowledge was due to geographical differences. Urban communities with limited access to resources could benefit from educational campaigns and support services that address their specific needs [52]. By acknowledging the community context, public health practitioners could develop more effective strategies to share important information about opioid overdose prevention and naloxone use [37,53,54]. On the other hand, the ICC for perceived competency was only 0.4%, which indicates that differences in perceived competency across zip codes are minimal, regardless of the availability of educational resources in these communities.

The findings of this study should be cautiously interpreted within the context of its limitations. By focusing on specific zip codes and recruiting survey respondents from only those zip codes, there may be limited generalizability, potentially

applying only to the settings with similar characteristics and environments. Nonetheless, the chosen zip codes were deliberately selected from urban areas with higher proportions of Black residents and elevated rates of overdose events and deaths compared to other areas in Indiana, rendering the findings of our study meaningful. Additionally, since the survey relied on self-reported data, it is susceptible to response bias and recall bias. Furthermore, we could not assess temporal relationships due to the cross-sectional nature of the survey. Also, the responses to opioid overdose items may have been subject to social desirability bias, potentially leading to overestimation of perceived competence. While our survey included 10 knowledge items and 7 competency items, these items may not have fully captured the breadth of understanding and skills needed for effective overdose response. Finally, the lack of qualitative data limits our ability to explore the underlying reasons behind observed disparities, such as structural barriers or cultural factors affecting knowledge and competency. Future research should incorporate qualitative approaches to better capture the nuances of opioid overdose knowledge and response competency.

This study aimed to assess the level of knowledge and perceived competency regarding opioid overdose and naloxone administration among urban Indiana residents and to examine racial and community-level differences. Our findings demonstrate that significant gaps in knowledge exist, particularly among Black residents living in high-poverty areas, underscoring the value of tailored education initiatives. While strategies to reduce opioid-involved deaths are multifaceted, ensuring layperson access to naloxone, improving knowledge of overdose recognition and response, and strengthening naloxone administration self-efficacy are critical components. As part of the MACRO-B project, we are currently providing community-based overdose prevention education, naloxone administration training, and naloxone distribution. Moving forward, we plan to conduct longitudinal and qualitative research to better understand the mechanisms underlying these differences and to evaluate the long-term effectiveness and sustainability of our intervention efforts.

## Supporting information

**S1 Fig. Survey administration timeline and follow-up procedure.**
(TIF)

**S1 Table. Demographic characteristics of survey respondents: March-May 2023 (N = 772).** Frequencies may not sum to the total and proportions may not sum to 100% due to missing observations.
(DOCX)

**S2 Table. Weighted frequencies and percentages for knowledge on opioid overdose and naloxone administration by race: March-May, 2023 (N = 772).** Frequencies may not sum to the total due to missing observations. Corresponding p-values for "a" were calculated via weighted adjusted Chi-square tests. Corresponding p-values for "b" were calculated via weighted design-based F-test.
(DOCX)

**S3 Table. Zip-code level predictors: 2022 American Community Survey 5-year estimates.** FPL = Federal Poverty Level. HS = high school. The "a" refers to the number of individuals with income below the 200 percent of federal poverty level. The "b" refers to the proportion of individuals with high school or higher educational attainment among those with 25 years or older. The "c" refers to the population size of each zip code.
(DOCX)

## Acknowledgments

This work was supported by the Indiana University Center for Survey Research (CSR). We thank the CSR for assisting in administering community surveys in our target study area which greatly assisted the research although they may not agree with all of the interpretations/conclusions of this paper.

## Author contributions

**Conceptualization:** Dong-Chul Seo.

**Formal analysis:** Shin Hyung Lee.

**Funding acquisition:** Dong-Chul Seo.

**Investigation:** Dong-Chul Seo.

**Methodology:** Shin Hyung Lee, Dong-Chul Seo.

**Supervision:** Dong-Chul Seo.

**Writing – original draft:** Shin Hyung Lee.

**Writing – review & editing:** Jon Agley, Vatsla Sharma, Francesca Williamson, Pengyue Zhang, Dong-Chul Seo.

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
