## [Decision Letter · Decision Letter 0]

Dear Dr. Seo,

Thank you for submitting your manuscript to PLOS ONE. After careful consideration, we feel that it has merit but does not fully meet PLOS ONE’s publication criteria as it currently stands. Therefore, we invite you to submit a revised version of the manuscript that addresses the points raised during the review process.

We look forward to receiving your revised manuscript.

Kind regards,

Junghwan Kim

Academic Editor

PLOS ONE

Journal Requirements:

“This project was supported to Dr. D.-C. Seo by the Office of Minority Health (OMH) of the U.S. Department of Health and Human Services (HHS) as part of a financial assistance award (CPIMP221346). For more information, please visit https://www.minorityhealth.hhs.gov/. The contents are those of the authors and do not necessarily represent the official views of, nor an endorsement, by OMH/OASH/HHS, or the U.S. Government.”

‘This work was supported by the Indiana University Center for Survey Research (CSR). We thank the CSR for assisting in administering community surveys in our target study area which greatly assisted the research although they may not agree with all of the interpretations/conclusions of this paper.”

“This project was supported to Dr. D.-C. Seo by the Office of Minority Health (OMH) of the U.S. Department of Health and Human Services (HHS) as part of a financial assistance award (CPIMP221346). For more information, please visit https://www.minorityhealth.hhs.gov/. The contents are those of the authors and do not necessarily represent the official views of, nor an endorsement, by OMH/OASH/HHS, or the U.S. Government.”

Reviewers' comments:

Reviewer's Responses to Questions

**Comments to the Author**

1. Is the manuscript technically sound, and do the data support the conclusions?

Reviewer #1: Yes

Reviewer #2: Partly

Reviewer #3: Yes

Reviewer #4: Yes

2. Has the statistical analysis been performed appropriately and rigorously?

Reviewer #1: Yes

Reviewer #2: I Don't Know

Reviewer #3: Yes

Reviewer #4: Yes

3. Have the authors made all data underlying the findings in their manuscript fully available?

Reviewer #1: Yes

Reviewer #2: Yes

Reviewer #3: No

Reviewer #4: Yes

4. Is the manuscript presented in an intelligible fashion and written in standard English?

Reviewer #1: Yes

Reviewer #2: Yes

Reviewer #3: Yes

Reviewer #4: Yes

Reviewer #1: 1- the response rate is relatively lo 23%

2- what are the steps taken to calculate demographic matching?

3- you used OOKS and OOAS without any clarifications for the psychometric validation?

4- There is a significant finding that Black participants in high-poverty areas had lower knowledge scores than their White . However, the mechanisms or contributing factors are not well explored

5- in limitation section you should mention social bias and lack of qualitative data and sing only 10 knowledge and 7 competency items

6- the method section should include response rate the different approach

7- table 2 is dense

8- add some diagrams or graphs

9- language and grammar editing

Reviewer #2: This article may be corrected as follows :

In abstract should be explained briefly related to IMRAD, furthermore in abstract also explained what becomes Novelty.

In the introduction, I have not found any Research Gap, so researchers need to improve; wherefrom the Research Gap, researchers can give something new from this research, and I have not found what is offered by researchers as Novelty.

Literature review explains how previous studies work, and you can tell what you find in previous research and what you find new from this research.

In the methodology may need to be explained in more detail related to the research subject so that it becomes clear who is the subject.

In results and discussion should give an idea of the basis of the research's purpose and discuss in-depth the findings in the field, and from there, researchers can give something new from this research.

please add Conclusions must certainly be able to answer from the purpose of this study.

*** I found the strengths of this article, but there are still weaknesses that need to be fixed.

This article is Acceptable in Plos One with Major Revision

Reviewer #3: Thank you for providing me the opportunity to review this manuscript. Overall I found this manuscript to be an interesting read, describing the disparity in knowledge of naloxone and opioid overdoses and be impacted by systemic issues related to race. It is a timely reminder that we all need to do more in our work to ensure the right messages are reaching the right people, regardless of race or other factors.

I think there are two areas of improvement needed for this manuscript: 1) being concise, and 2) clearly describing all statistical analyses completed.

I am a firm believer of using less words where possible, and I would suggest the authors look at areas where they could reduce the word count. At times, some sections felt overly described. Perhaps the section that stood out to me the most was when describing listwise deletion and why it was used. I would assume most people reading would understand listwise deletion, and that it's use when describing could be said it in

fewer words. As another suggestion to shorten the methods, is provide the survey instrument as a supplementary to reduce the word count in the methods when describing the survey. I appreciate the authors have submitted to a non-addiction or drug and alcohol journal, and therefore at times need to describe certain concepts in greater details.

I was confused as the methods made no mention of using chi-square or t-test analyses, and yet table 1 and table 2 have results from these tests. It was not until i noticed the footnotes of each table that i realized these tests were completed for each table. It would be important to ensure this is adequately described within the statistical analyses section. Did the authors utilize a Bonferroni correction for the chi-square test? We're there further analyses completed that as a reader I am not aware of?

Otherwise, no major issues with the content of this manuscript. A few minor typos, but otherwise a sound paper that requires some finessing.

Reviewer #4: For author and editor,

Review of the submission #PONE-D-24-50788: Opioid Overdose and Naloxone Administration Knowledge and Perceived Competency in A Probability Sample of Indiana Urban Communities with Large Black Populations". This Original Research Report is in accordance with focus and scope of PLOS One. The manuscript is related to an original theme and relevant to the area. Considerations, corrections and suggestions are in the attached file.

Considering the scientific merit and originality of the manuscript, the recommendation is: “MINOR REVISION”.

Cordially.

**Do you want your identity to be public for this peer review?** For information about this choice, including consent withdrawal, please see our Privacy Policy

Reviewer #1: No

Reviewer #2: **Yes: ** Imaduddin Hamzah

Reviewer #3: No

Reviewer #4: **Yes: ** Pablo Guilherme Caldarelli

---

## [Author Response · Author response to Decision Letter 1]

14 May 2025

Editor:

1. Please ensure that your manuscript meets PLOS ONE's style requirements, including those for file naming. The PLOS ONE style templates can be found here:

We have edited to follow the style requirements.

2. Please provide an amended statement that declares *all* the funding or sources of support (whether external or internal to your organization) received during this study, as detailed online in our guide for authors at http://journals.plos.org/plosone/s/submit-now. Please also include the statement “There was no additional external funding received for this study.” in your updated Funding Statement. Please include your amended Funding Statement within your cover letter. We will change the online submission form on your behalf.

Done as requested.

3. We note that you have indicated that there are restrictions to data sharing for this study. For studies involving human research participant data or other sensitive data, we encourage authors to share de-identified or anonymized data. However, when data cannot be publicly shared for ethical reasons, we allow authors to make their data sets available upon request. For information on unacceptable data access restrictions, please see http://journals.plos.org/plosone/s/data-availability#loc-unacceptable-data-access-restrictions. Before we proceed with your manuscript, please address the following prompts: If there are ethical or legal restrictions on sharing a de-identified data set, please explain them in detail (e.g., data contain potentially identifying or sensitive patient information, data are owned by a third-party organization, etc.) and who has imposed them (e.g., a Research Ethics Committee or Institutional Review Board, etc.). Please also provide contact information for a data access committee, ethics committee, or other institutional body to which data requests may be sent. Please update your Data Availability statement in the submission form accordingly.

We will make datasets available upon request. We have updated Data Availability Statement as requested.

4. Please remove any funding-related text from the manuscript and let us know how you would like to update your Funding Statement.

In response to Request #2 above, we have already amended our Funding Statement and included it within our revised cover letter. We have removed all funding-related text from the abstract and main narrative.

Done as requested.

Reviewer #1:

The response rate is relatively low 23%

While the overall response rate of 23% is relatively low, it is consistent with the rates typically reported in contemporary household-based probability surveys using mail and web modes without in-person follow-up. Recent reports from the American Association for Public Opinion Research (AAPOR) attribute declining response rates to broader trends such as survey fatigue and reduced willingness to participate. To minimize potential nonresponse bias, we employed probability sampling, survey weighting, and multiple follow-up strategies, including reminder letters, mailed paper surveys, and incentives. Furthermore, the demographic composition of our final analytic sample closely matched census benchmarks for the target communities, suggesting limited risk of nonresponse bias.

This is also supported by the findings from a replicated methodological study by Keeter and his colleagues (Keeter et al., 2006) which compared standard recruitment (25% response rate) with a rigorous recruitment (50% response rate) using random-digit-dial (RDD) methods. Across 77 of 84 variables, the surveys produced statistically indistinguishable results, indicating that unit nonresponse within this response rate range did not meaningfully compromise data quality (Keeter et al., 2006). The study also found that even respondents hardest to reach or those who partially completed interviews did not differ systematically in key behaviors or attitudes (Keeter et al., 2006) - further supporting the robustness of survey estimates despite modest response rates. [Keeter, S., Kennedy, C., Dimock, M., Best, J., & Craighill, P. (2006). Gauging the impact of growing nonresponse on estimates from a national RDD telephone survey. International Journal of Public Opinion Quarterly, 70(5), 759-779]

What are the steps taken to calculate demographic matching?

We used four zip-code level demographic matching variables: Black racial proportion, families living in poverty, population size, and proportion of children eligible for free/reduced lunch. Detailed information has been provided in text (Lines 144 – 152).

You used OOKS and OOAS without any clarifications for the psychometric validation?

We apologize for the confusion. While OOKS (on knowledge) and OOAS (on attitudes) are companion scales, we only used OOKS to measure knowledge. As requested, we have computed and reported the psychometric properties of both the knowledge scale and the perceived level of competency in managing opioid overdoses scale in our sample. The Cronbach’s alpha for the perceived level of competency was 0.87 but it was 0.41 for the knowledge scale. Although the Cronbach’s alpha in our sample was modest for the knowledge scale, we have provided justification for its acceptability by citing two supporting articles. Specifically, one should not expect a knowledge test to function in a similar psychometric manner to a scale intended to measure a singular construct, since not all knowledge items measure the same constructs. We have revised the text accordingly (Lines 165 – 170).

There is a significant finding that Black participants in high-poverty areas had lower knowledge scores than their White counterparts. However, the mechanisms or contributing factors are not well explored.

We appreciate the comments. We agree that the mechanisms underlying the observed differences are important to explore. However, as this study represents baseline, exploratory findings within a larger intervention project, no prior analyses among Indiana urban residents were available to further investigate potential contributing factors. We have clarified this in the revised manuscript and highlighted that while the results suggest important differences, identifying specific mechanisms will require further longitudinal and qualitative research (Lines 365 – 368).

In limitation section you should mention social bias and lack of qualitative data and using only 10 knowledge and 7 competency items

As requested, we have added the limitation of potential social desirability bias and the limited number of knowledge and competency items, and the absence of qualitative data (Lines 361 – 368).

The method section should include response rate the different approach

As requested, we have revised the Methods section to explicitly report the overall response rate (23%) and added a detailed description of the sequential survey administration approach, including mailings, reminders, and incentive use (Lines 153 – 162). To further clarify the data collection process, we have also created and included a flowchart (Figure S2) that visually summarizes the survey administration timeline and participation outcomes.

Table 2 is dense

As requested, we have revised Table 2 to streamline its content and improve readability. Additionally, we have moved specific survey items to Table S4 to make the main table more concise while still providing detailed information for readers who wish to review it.

Add some diagrams or graphs

Done as suggested. As mentioned above, we have developed a flowchart illustrating the sample selection and survey administration process to enhance clarity. The flowchart has been added as Figure S2 in the revised manuscript.

Language and grammar editing

Done as suggested.

Reviewer #2:

In abstract should be explained briefly related to IMRAD, furthermore in abstract also explained what becomes Novelty.

As requested, we have revised the Abstract to follow the IMRAD structure more clearly, providing distinct sections for the background, methods, results, and conclusions. In addition, we have added a sentence highlighting the novelty of the study, specifically noting how our findings contribute to new evidence regarding the interaction between structural factors (e.g., community poverty) and race in shaping opioid overdose knowledge (Lines 42 – 62).

In the introduction, I have not found any Research Gap, so researchers need to improve; where from the Research Gap, researchers can give something new from this research, and I have not found what is offered by researchers as Novelty. Literature review explains how previous studies work, and you can tell what you find in previous research and what you find new from this research.

As requested, we have revised the Introduction and literature review section to clarify the research gap and the novelty of the study (Lines 84 – 86 & 111 – 118).

In the methodology may need to be explained in more detail related to the research subject so that it becomes clear who is the subject.

Done as suggested (Lines 144 – 152)

In results and discussion should give an idea of the basis of the research's purpose and discuss in-depth the findings in the field, and from there, researchers can give something new from this research.

As requested, we have revised the opening section of the Discussion section to better connect our findings back to the study’s research purpose, emphasize the novelty of our work, and more clearly explain how our findings contribute new insights to the field (Lines 286 – 298).

Please add Conclusions must certainly be able to answer from the purpose of this study.

Done as suggested (Lines 369 – 380)

Reviewer #3:

Thank you for providing me the opportunity to review this manuscript. Overall I found this manuscript to be an interesting read, describing the disparity in knowledge of naloxone and opioid overdoses and be impacted by systemic issues related to race. It is a timely reminder that we all need to do more in our work to ensure the right messages are reaching the right people, regardless of race or other factors.

Thank you for your thoughtful and encouraging feedback.

I think there are two areas of improvement needed for this manuscript: 1) being concise, and 2) clearly describing all statistical analyses completed.

In view of the reviewer comments, we have revised the manuscript to be more concise and have clarified the description of all statistical analyses.

I am a firm believer of using less words where possible, and I would suggest the authors look at areas where they could reduce the word count. At times, some sections felt overly described. Perhaps the section that stood out to me the most was when describing listwise deletion and why it was used. I would assume most people reading would understand listwise deletion, and that it's use when describing could be said it in fewer words. As another suggestion to shorten the methods, is provide the survey instrument as a supplementary to reduce the word count in the methods when describing the survey. I appreciate the authors have submitted to a non-addiction or drug and alcohol journal, and therefore at times need to describe certain concepts in greater details.

As requested, we have revised the manuscript to make it concise where possible, including the explanation of listwise deletion. Additionally, we have moved the detailed description of the survey procedure to Figure S2, as suggested.

I was confused as the methods made no mention of using chi-square or t-test analyses, and yet table 1 and table 2 have results from these tests. It was not until i noticed the footnotes of each table that i realized these tests were completed for each table. It would be important to ensure this is adequately described within the statistical analyses section. Did the authors utilize a Bonferroni correction for the chi-square test? Are there further analyses completed that as a reader I am not aware of?

We apologize for the lack of clarity in the Methods section. We have revised the Statistical Analyses subsection to explicitly state that we conducted adjusted chi-square tests with design-based Rao-Scott corrections for categorical variables and design-based F-tests for continuous variables in Tables 1 and 2 (Lines 214 – 217). No Bonferroni correction was applied for these descriptive comparisons for two reasons. First, even with Bonferroni adjustment with 10 pairwise comparisons (using 0.005 rather than 0.05 as the p value threshold), all the reported statistical significance stays unchanged. Second, these analyses were intended to characterize the (sub)samples rather than hypotheses-driven statistical tests.

Reviewer #4:

The abstract is intelligible and accurately describes the objective and results obtained. It is a sufficient summary of the contents of the paper. In the abstract, the results should be explored. The CONCLUSIONS could be directly related to the results found in the study.

Thank you for the thoughtful feedback. As suggested, we have revised the abstract to explain the results and to ensure the conclusions are directly tied to the findings of the study.

METHODS: The methods were described and organized in sufficient detail. However, the chapter is quite long. I would suggest to the authors to summarize it. A suggestion would be to present another flowchart with the methodological organization.

As suggested, we have summarized the methods section to improve clarity and conciseness and have included an additional flowchart in Figure S2.

RESULTS: The results are interesting. The authors clearly highlighted the information collected using the methods described to meet the study objectives. The tables with the presentation of the results are organized and clear. The authors identify the limitations of the study in the end of the manuscript, but it is necessary to indicate suggestions for future research.

As suggested, we have provided suggestions for future research (Lines 378 – 380).

CONCLUSION. The authors could present a clear and objective conclusion.

As suggested, we have revised the Conclusion section to present a clearer and more objective summary of the findings (Lines 369 – 380).

REFERENCES: The authors cite appropriate and actual papers for comments/discussion made

Thank you.

---

## [Decision Letter · Decision Letter 1]

Dear Dr. Seo,

Thank you for submitting your manuscript to PLOS ONE. After careful consideration, we feel that it has merit but does not fully meet PLOS ONE’s publication criteria as it currently stands. Therefore, we invite you to submit a revised version of the manuscript that addresses the points raised during the review process.

We look forward to receiving your revised manuscript.

Kind regards,

Junghwan Kim

Academic Editor

PLOS ONE

Journal Requirements:

Reviewers' comments:

Reviewer's Responses to Questions

**Comments to the Author**

Reviewer #1: (No Response)

Reviewer #2: (No Response)

Reviewer #3: All comments have been addressed

Reviewer #4: All comments have been addressed

2. Is the manuscript technically sound, and do the data support the conclusions?

Reviewer #1: Yes

Reviewer #2: Yes

Reviewer #3: Yes

Reviewer #4: Yes

3. Has the statistical analysis been performed appropriately and rigorously?

Reviewer #1: Yes

Reviewer #2: Yes

Reviewer #3: Yes

Reviewer #4: Yes

4. Have the authors made all data underlying the findings in their manuscript fully available?

Reviewer #1: Yes

Reviewer #2: Yes

Reviewer #3: Yes

Reviewer #4: Yes

5. Is the manuscript presented in an intelligible fashion and written in standard English?

Reviewer #1: Yes

Reviewer #2: Yes

Reviewer #3: Yes

Reviewer #4: Yes

Reviewer #1: 1- Section 2.2, paragraph 1.explain how these areas were chosen beyond demographic criteria

2- Section 2.3.3. ...defined by whether individual had personally experienced an opioid overdose or had a family member or close friend who died

3- Section 2.3.3, Table 1. confusing

4- Tables 3 and 4, Results narrative

5- Results sections rely only on tables

6- Discussion paragraphs 1 and 2 closely mirror Results

7- Abstract is dense and underrepresents key findings in plain language.

8- Citations 5, 7, 40 used heavily

9- No discussion of studies that show minimal or no racial difference in naloxone administration

10- No figures

11- Table captions could better explain

12-

Reviewer #2: This article may be corrected as follows :

Please look again at the author's guidelines on abstract, where abstract is mapped with background, objective, result and conclusion. A good abstract already looks what is novelty in this article.

In the introduction, I have not found any Research Gap, so researchers need to improve; wherefrom the Research Gap, researchers can give something new from this research, and I have found what is offered by researchers as Novelty.

In results and discussion should give an idea of the basis of the research's purpose and discuss in-depth the findings in the field, and from there, researchers can give something new from this research.

*** I found the strengths of this article, but there are still weaknesses that need to be fixed.

This article is Acceptable with Minor Revision in Plos Ones Journal

Reviewer #3: The authors have addressed comments by reviewers sufficiently and I have no further suggestions for this manuscript.

Reviewer #4: For authors and Editor,

Review of the submission ID PONE-D-24-50788R1: “Opioid Overdose and Naloxone Administration Knowledge and Perceived Competency in A Probability Sample of Indiana Urban Communities with Large Black Populations”

Thank you very much for submitting the manuscript to the PLOS One.

The revisions that you made to the manuscript are very effective in addressing the remaining concerns. Following the review of the previous version, considering the scientific merit, the originality of the manuscript and the reading of the new version, the recommendation is: "ACCEPT".

Congratulation one your fine research!

Cordially,

**Do you want your identity to be public for this peer review?** For information about this choice, including consent withdrawal, please see our Privacy Policy

Reviewer #1: **Yes: ** Yousra Nomeir

Reviewer #2: **Yes: ** ok

Reviewer #3: No

Reviewer #4: **Yes: ** Pablo Guilherme Caldarelli

---

## [Author Response · Author response to Decision Letter 2]

30 Jun 2025

Reviewer #1:

1. Section 2.2, paragraph 1.explain how these areas were chosen beyond demographic criteria

Revised to include additional details on area selection (Lines 146-157, 160-161).

2. Section 2.3.3. ...defined by whether individual had personally experienced an opioid overdose or had a family member or close friend who died

Revised to clarify the construction of the opioid overdose history variable. Specifically, the binary variable was based on responses to two survey items: (1) whether the respondent had personally experienced an opioid overdose and (2) whether a family member or close friend had died from an opioid overdose (Lines 208-212).

3. Section 2.3.3, Table 1. Confusing

Revised the table notes to clarify the interpretation of numbers and how the opioid overdose history variable was constructed (Lines 559-565).

4. Tables 3 and 4, Results narrative - Results sections rely only on tables

Expanded the results narrative to integrate key findings from tables (Lines 263-265, 282-283, 296-298).

5. Discussion paragraphs 1 and 2 closely mirror Results

Revised the discussion to reduce overlap and enrich interpretations (Lines 303-305, 310-312, 325-328).

6. Abstract is dense and underrepresents key findings in plain language.

Revised for clarity and a better representation of main findings (Lines 42-50, 60-63).

7. Citations 5, 7, 40 used heavily

Citations 5 and 40 are foundational but are now balanced with additional supporting references.

8. No discussion of studies that show minimal or no racial difference in naloxone administration

Addressed with literature and integrated into the discussion (Lines 318-322)

9. No figures

A figure in the supplementary document has been created using PACE digital diagnostic tool and uploaded.

10. Table captions could better explain

Revised as suggested (Lines 559-565).

Reviewer #2:

1. This article may be corrected as follows:

Please look again at the author's guidelines on abstract, where abstract is mapped with background, objective, result and conclusion. A good abstract already looks what is novelty in this article.

Revised to align with the structured format and to emphasize novelty of the study.

2. In the introduction, I have not found any Research Gap, so researchers need to improve; wherefrom the Research Gap, researchers can give something new from this research, and I have found what is offered by researchers as Novelty. In results and discussion should give an idea of the basis of the research's purpose and discuss in-depth the findings in the field, and from there, researchers can give something new from this research.

Revised the introduction to clarify the research gap and articulate the study’s novelty (Lines 112-122). The discussion has also been expanded to better connect with the study aims (Lines 303-305, 310-312, 318-322, 325-328).

Reviewer #3:

1. The authors have addressed comments by reviewers sufficiently and I have no further suggestions for this manuscript.

Thank you.

Reviewer #4:

The revisions that you made to the manuscript are very effective in addressing the remaining concerns. Following the review of the previous version, considering the scientific merit, the originality of the manuscript and the reading of the new version, the recommendation is: "ACCEPT". Congratulation one your fine research!

Thank you.

---

## [Editor Report · Decision Letter 2]

Opioid Overdose and Naloxone Administration Knowledge and Perceived Competency in A Probability Sample of Indiana Urban Communities with Large Black Populations

PONE-D-24-50788R2

Dear Dr. Seo,

We’re pleased to inform you that your manuscript has been judged scientifically suitable for publication and will be formally accepted for publication once it meets all outstanding technical requirements.

Kind regards,

Junghwan Kim

Academic Editor

PLOS ONE
---

## [Editor Report · Acceptance letter]

PONE-D-24-50788R2

PLOS ONE

Dear Dr. Seo,

I'm pleased to inform you that your manuscript has been deemed suitable for publication in PLOS ONE. Congratulations! Your manuscript is now being handed over to our production team.

Kind regards,

on behalf of

Dr. Junghwan Kim

Academic Editor

PLOS ONE